# The Seroprevalence of SARS-CoV-2 IgG Antibodies in Children Hospitalized for Reasons Other Than COVID-19

**DOI:** 10.3390/jcm11133819

**Published:** 2022-07-01

**Authors:** Justyna Franczak, Justyna Moppert, Małgorzata Sobolewska-Pilarczyk, Małgorzata Pawłowska

**Affiliations:** 1Department of Paediatrics, Infectious Diseases and Hepatology, Voivodeship Infectious Observation Hospital in Bydgoszcz, 85-030 Bydgoszcz, Poland; korzybska.justyna@gmail.com (J.F.); m.pilarczyk@wsoz.pl (M.S.-P.); mpawlowska@cm.umk.pl (M.P.); 2Department of Infectious Diseases and Hepatology, Collegium Medicum in Bydgoszcz, Nicolaus Copernicus University in Toruń, 87-100 Toruń, Poland

**Keywords:** COVID-19, SARS-CoV-2, antibody, children

## Abstract

The aim of the study was to assess the seroprevalence of SARS-CoV-2 IgG antibodies in children hospitalized for reasons other than COVID-19. The study was conducted among 686 children, aged 2 weeks to 18 years, hospitalized in the Department of Paediatrics, Infectious Diseases, and Hepatology in Bydgoszcz, Poland, in the period from 1 June 2021 to 30 April 2022. The presence of anti-SARS-CoV-2 antibodies was detected in 392 (57%) children. Since December 2021, a significantly larger proportion of children with a positive titer of anti-SARS-CoV-2 antibodies has been observed, constituting as much as 87.5% of patients hospitalized in April 2022. In total, 69% of children with detected anti-SARS-CoV-2 antibodies were children under 5 years of age. The highest mean amounts of titers of anti-SARS-CoV-2 antibodies were observed in children over 10 years of age. The conducted studies showed the presence of anti-SARS-CoV-2 antibodies in a significant group of hospitalized children and an increase in the percentage of these children during the fourth and fifth wave of COVID-19 in Poland caused by the Delta and Omicron variants, respectively. The vast majority of parents of the studied children had no knowledge of the COVID-19 infection in their charges, which may indicate asymptomatic infection or mild course of the disease.

## 1. Introduction

COVID-19 is an acute infectious disease caused by the SARS-CoV-2 virus. The first case was reported in November 2019 in Wuhan City, Hubei Province, Central China. The virus turned out to be highly contagious and quickly spread to many countries around the world, taking the dimensions of a pandemic announced by the WHO on 11 March 2020 [1,2]. 

During the course of the pandemic, different variants of the SARS-CoV-2 virus appeared, mainly related to changes in the RBD domain of the spike protein, and these variants were: Alpha (B.1.1.7), Beta (B.1.351), Gamma (P.1), Delta (B.1.617.2), and Omicron (B.1.1.529) [3]. SARS-CoV-2 is spread mainly by airborne droplets or dust particles through the respiratory tract and by direct contact of mucous membranes with contaminated surfaces [4].

COVID-19 occurs in various clinical stages: asymptomatic or mild, stable, with respiratory and/or systemic symptoms, unstable stage of respiratory failure, and critical condition with acute respiratory distress syndrome (ARDS) [5,6]. Risk factors for severe disease and death are older age and comorbidities, such as obesity, diabetes, hypertension, coronary artery disease, chronic obstructive pulmonary disease, chronic kidney disease, cancer, and immune disorders [7]. According to world data, 80% of patients have no significant clinical symptoms or they are mild. Symptoms of severe interstitial pneumonia of varying severity predominate in about 20% of those infected [8]. 

Children of all ages can get COVID-19. The course of infection in this group is usually mild, and the most common symptoms are fever and respiratory ailments, e.g., runny nose, dry cough, or pharyngitis. Patients also report diarrhea, abdominal pain, vomiting, fatigue, headache, loss of smell and taste, muscle aches, and shortness of breath [9,10]. About 15% of infected children develop severe COVID-19 with pneumonia, while 5% of patients develop thrombosis, septic shock, and multiorgan failure due to a cytokine storm resulting from an abnormal immune system response [11]. The group of children at risk of severe COVID-19 is made up of newborns, children with obesity, and coexisting chronic diseases (such as congenital diseases of the heart, lungs, and respiratory tract, malnutrition, and cancer) [12,13].

The aim of the study was to assess the seroprevalence of SARS-CoV-2 IgG antibodies in children hospitalized for reasons other than COVID-19.

## 2. Materials and Methods

The study was conducted among 686 children, aged 2 weeks to 18 years, hospitalized in the Department of Paediatrics, Infectious Diseases, and Hepatology of the Voivodeship Infectious Observation Hospital in Bydgoszcz, Poland, in the period from 1 June 2021 to 30 April 2022. The reasons for hospitalization were acute gastroenteritis, respiratory infections, hepatitis of various etiologies, infectious mononucleosis, chicken pox, and other infectious diseases. All study patients tested negative for SARS-CoV-2 antigen/nucleid acid. During their stay in the hospital, serum was collected from the patients in order to perform a serological test for SARS-CoV-2 infection. A short survey was also conducted among the patients and caregivers regarding COVID-19 disease and coexisting chronic diseases. Children vaccinated against SARS-CoV-2 or hospitalized due to COVID-19 were excluded from the study. The presence of anti-SARS-CoV-2 antibodies was determined by the LIAISON SARS-CoV-2 TrimericS—IgG test using the chemiluminescence method to quantify the anti-Trimer-S (Spike) protein IgG antibodies of the SARS-CoV-2 virus.

This study was approved by Bioethics Committee of the Nicolaus Copernicus University in Torun, Collegium Medicum in Bydgoszcz. Written informed consent was obtained from the parents of all study children on admission to the hospital. 

## 3. Results

Anti-SARS-CoV-2 antibodies were detected in 392 (57%) of the 686 study children. Among these antibody positive children, 219 (56%) were boys. From December 2021, a significant predominance of patients with positive p/SARS-CoV-2 antibody titers was observed, reaching 87.5% of children hospitalized in April 2022 (Figure 1).

The presence of detectable anti-SARS-CoV-2 antibodies was more often found in younger age groups. A total of 36% (141 people) were children from 3 to 5 years old, 33.4% (131 people) were children under 2 years of age, 19.1% (75 people) were children from 6 to 10 years of age, and 11.5% (45 people) were children from 11 to 18 years of age. The median age was 3.5 years.

Among the 392 patients positive for anti-SARS-CoV-2 antibodies, only 47 (12%) reported the laboratory-confirmed COVID-19 infection in the past. A history of chronic disease or specialist care was found in 86 people (22%). The most common of these diseases were allergic diseases (in 50 people): bronchial asthma in 13 people, inhalation allergy in 13 people, atopic dermatitis in 13 people, and food allergy in 11 people. Parents of patients with chronic diseases were slightly more aware of COVID-19 disease in their children than parents of patients without (16% vs. 11%).

In February, March, and April 2022, the mean titers of anti-SARS-CoV-2 antibodies reached higher values compared to the previous months (Figure 2). Higher values of the antibody titer were also noticed in children over 10 years of age. The mean titers of antibodies depending on the age of the patients are presented in Table 1.

## 4. Discussion

A study conducted in the Department of Paediatrics, Infectious Diseases, and Hepatology of the Voivodeship Infectious Observation Hospital in Bydgoszcz confirms that the SARS-CoV-2 virus is also significantly spreading among children. In June 2021, and in September, October, and November 2021, the presence of anti-SARS-CoV-2 antibodies, evidence of infection, was found in about 40% of children hospitalized in this hospital for reasons other than COVID-19. The exception was the summer months—July and August 2021—in which positive antibodies were found in 61% and 53% of children, respectively. These data could result from a random group or reflect the spring wave of the disease. However, from December 2021, a systematic increase in the percentage of children with anti-SARS-CoV-2 antibodies was found, which, in April 2022, accounted for 87.5% of patients (Figure 1). Our findings are consistent with a noticeable increase in childhood morbidity during the fourth and fifth waves of COVID-19 in Poland, caused by the Delta and Omicron variants, respectively.

The majority of children with anti-SARS-CoV-2 antibodies detected (272 patients—69%) were children under 5 years of age. This age range may be due to the fact that younger children were hospitalized more often, which made them more frequent participants in the study. In addition, they were more prone to falling ill because they did not comply with the sanitary regime (e.g., they did not wear masks), they attended nurseries and kindergartens, which operated even at the peak of the disease outbreak, and they were not covered by SARS-CoV-2 vaccinations. Furthermore, an immature immune system in young children is associated with a higher risk of infectious diseases.

The vast majority of parents of the studied children denied that their charges had a COVID-19 infection, which indicates a poorly symptomatic or mild course of the disease that does not require diagnostics. Additionally, a reluctance of parents to test their children was observed, resulting from the discomfort for the child of nasopharyngeal swab collection. Caregivers of patients with chronic diseases were more often aware of COVID-19 in their children than caregivers of patients without. The children suffering from chronic diseases more often used hospital care than the healthy children, which undoubtedly translated into more frequent testing, e.g., before planned hospitalization, rehabilitation camps or diagnostic tests, and incidental diagnosis of SARS-CoV-2 infection. The children in this group also had a higher risk of a severe course of infection requiring hospitalization and extended diagnostics.

In February and March 2022, the mean titers of anti-SARS-CoV-2 antibodies reached higher values compared to the previous months, which was probably related to different variants of the SARS-CoV-2 virus. Additionally, older children had higher antibody titers than younger patients.

In the literature, there are few studies on the presence of anti-SARS-CoV-2 antibodies in children. In a meta-analysis, Bhuiyan et al. found that 50% of COVID-19 infected were infants and 53% of the children were male. The course of infection was mild and moderate in >90% of the patients, and the course was asymptomatic in 43% of the children [14]. In contrast, Forrest et al. in their study concluded that 66.4% children had asymptomatic infection, 26.9% had mild symptoms, and 4.6% had moderate symptoms. The risk factors for the severity of the disease were chronic conditions and young age (infants) [15]. Marks et al., in their analysis, noted the number of hospitalizations of children infected with SARS-CoV-2 during the dominance of the Omicron variant, especially in the 0–6 month age group (which accounted for 44% of patients). It was found that 63% of the children had no comorbidities [16]. Sobolewska-Pilarczyk et al. published an analysis of the clinical course of COVID-19 in 300 infants registered in the multi-center SARSTerPED database. During the first wave of the pandemic (March to August 2020), 10.5% of the children diagnosed were infants, and during the second wave (September to December 2020), it was 30.7%. COVID-19 in the infants tested usually manifested as a mild infection of the gastrointestinal or respiratory tract. The most frequently observed symptoms were fever, cough, and runny nose. Pneumonia was diagnosed in 23% of the children. Co-morbidities, including birth defects, epilepsy, prematurity, atopic dermatitis, bronchopulmonary dysplasia, and immunodeficiency, were reported in 12% of the hospitalized infants. According to the authors, the actual incidence of COVID-19 in children is underestimated due to the lack of diagnoses of mild and asymptomatic courses [11].

## 5. Conclusions

The conducted research showed the presence of anti-SARS-CoV-2 antibodies in a significant group of children hospitalized for reasons other than COVID-19 and an increase in the percentage of such children during the fourth and fifth wave of COVID-19 in Poland caused by the Delta and Omicron variants, respectively. The vast majority of parents of the studied children did not know about the COVID-19 infection in their charges, which indicates a poorly symptomatic or mild course of the disease.

## Figures and Tables

**Figure 1 jcm-11-03819-f001:**
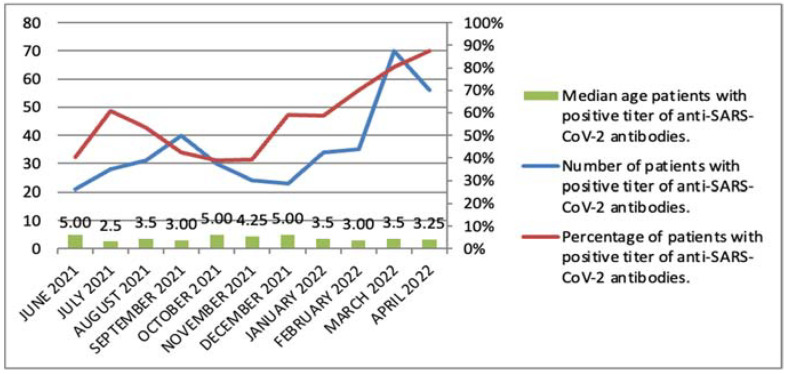
Median age, number, and percentage of patients with positive titer of anti-SARS-CoV-2 antibodies.

**Figure 2 jcm-11-03819-f002:**
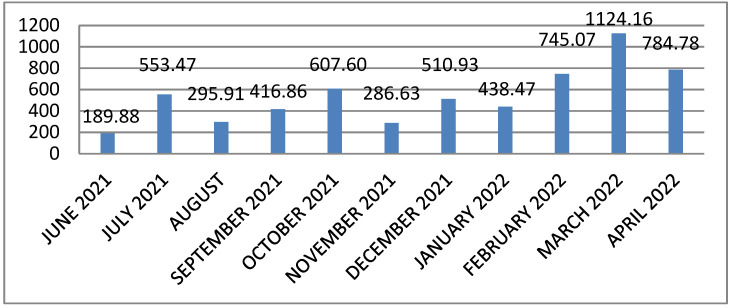
Mean titers of antibodies (Bau/mL) depending on the month of onset.

**Table 1 jcm-11-03819-t001:** Median and mean titers of anti-SARS-CoV-2 antibodies depending on age.

Age	Median Titers of Anti-SARS-CoV-2 Antibodies (Bau/mL)	Mean Titers of Anti-SARS-CoV-2 Antibodies (Bau/mL)
0–1 years	252.00	437.04
2–5 years	350.00	642.65
6–10 years	253.80	537.19
11–18 years	345.80	948.74

## Data Availability

All data generated and analysed during this study are included in this article.

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
