# Peer review of "The Seroprevalence of SARS-CoV-2 IgG Antibodies in Children Hospitalized for Reasons Other Than COVID-19"

_jcm, 2022, doi:10.3390/jcm11133819_

Round 1

Reviewer 1 Report

Journal: Journal of Clinical Medicine

Manuscript ID: jcm-1794574

Title: The frequency of SARS-CoV-2 infection in children hospitalized for reasons other than COVID-19.

Summary/General Reviewer Comments:  The manuscript describes the results of a serosurvey

for SARS-CoV-2 IgG antibodies among 686 children hospitalized for non-COVID-19 reasons.  The authors identified a high prevalence of antibodies in the children, with seroprevalence varying by age group and with an increasing prevalence trend from Dec, 2021 through Apr, 2022, consistent with the dissemination of new virus variants.  These findings are confirmatory, but nevertheless a valuable contribution to our understanding of SARS-CoV-2 infections in non-COVID-19 children.  A significant concern of this reviewer is the lack of any statement indicating that study approval was obtained from appropriate institutional review boards or parents/guardians of participating student subjects. 

Specific Reviewer Comments:

1)      The manuscript could benefit from overall shortening.  Specifically, Lines 32-36 and Lines 60-81 could be eliminated or dramatically shortened.  The Introduction should conclude with the study purpose/goals, not a review of the EU approved vaccines that do not relate to the study.  The detailed summaries of other study citations in the Discussion [Lines 164-197] should be retained, but shortened and condensed into a single paragraph.  Figures 2 and 3 could be removed as they are redundant with the text.  Figures 1 and 4 could be consolidated into a single Figure.

2)      Abstract, Line 23. Change to, “… which may indicate asymptomatic infection or mild course of the disease.”

3)      Materials and Methods, Line 84.  The hospitalized study population should be better defined.  What were they hospitalized for and how were they enrolled to avoid selection bias? 

4)      Materials and Methods, Line 91.  Replace with, “the LIAISON SARS-CoV-2 TrimericS-IgG test”.

5)      Results, Lines 95-97.  Replace with, “Anti-SARS-CoV-2 antibodies were detected in 392 (57%) of the 686 study children. Among these antibody positive children, 219 (56%) were boys.” 

6)      Figures 1 and 4.  To interpret these Figures, readers should be provided with the number (N) of patients at each time point (month).  Ideally the median age of each bin should also be  analyzed as age correlates with antibody detection/titer and confound comparative proportions and trends.

7)      Figure 4 and Table 1.  Median values might be a better representation of the average values than Means as they are less susceptible of extreme values.

8)      Results, Line 103.  Replace with, “The presence of detectable anti-SARS-CoV-2 antibodies …”.

9)      Results, Lines 110-113.  Replace with, “Among the 392 patients positive for anti-SARS-CoV-2 antibodies, only 47 (12%) reported COVID-19 symptoms.” What time period preceding enrollment were participants  asked to recall having COVID-19 symptoms?

10)  Discussion, Lines 138-140.  Replace with, “Our findings are consistent with a noticeable increase in childhood morbidity during the fourth and fifth waves of COVID-19 in Poland, …”. 

Author Response

Response to Reviewer 1 Comments

1)      The manuscript could benefit from overall shortening.  Specifically, Lines 32-36 and Lines 60-81 could be eliminated or dramatically shortened.  The Introduction should conclude with the study purpose/goals, not a review of the EU approved vaccines that do not relate to the study.  The detailed summaries of other study citations in the Discussion [Lines 164-197] should be retained, but shortened and condensed into a single paragraph.  Figures 2 and 3 could be removed as they are redundant with the text.  Figures 1 and 4 could be consolidated into a single Figure.

Response 1: The Lines 32-36 i 60-81 have been eliminated.

The Introduction was concluded with the study purpose. :

The aim of the study was to assess the seroprevalence of SARS-CoV-2 IgG antibodies in children hospitalized for reasons other than COVID-19.

Figures 2 and 3 have been removed.

Figures 1 and 4 haven't been removed, because it seems more readable to present the data in 2 figures in the author's opinion.

 The detailed summaries of other study citations in the Discussion [Lines 164-197] have been shortened and condensed into a single paragraph.

2)      Abstract, Line 23. Change to, “… which may indicate asymptomatic infection or mild course of the disease.”

Response 2: This Line has been changed.

The vast majority of parents of the studied children had no knowledge of the COVID-19 infection in their charges, which may indicate asymptomatic infection or mild course of the disease.

3)      Materials and Methods, Line 84.  The hospitalized study population should be better defined.  What were they hospitalized for and how were they enrolled to avoid selection bias? 

Response 3:

The reasons for hospitalization were acute gastroenteritis, respiratory infections, hepatitis of various etiologies, infectious mononucleosis, chicken pox, and other infectious diseases.

Children vaccinated against SARS-CoV-2 or hospitalized due to COVID-19 were excluded from the study.

4)      Materials and Methods, Line 91.  Replace with, “the LIAISON SARS-CoV-2 TrimericS-IgG test”.

Response 4: This Line has been changed.

The presence of anti-SARS CoV-2 antibodies was determined by the LIAISON SARS-CoV-2 TrimericS-IgG test using the chemiluminescence method to quantify the anti-Trimer-S (Spike) protein IgG antibodies of the SARS-CoV-2 virus.

5)      Results, Lines 95-97.  Replace with, “Anti-SARS-CoV-2 antibodies were detected in 392 (57%) of the 686 study children. Among these antibody positive children, 219 (56%) were boys.” 

Response 5: The Line 95-97 has been replaced.

      Anti-SARS-CoV-2 antibodies were detected in 392 (57%) of the 686 study children. Among these antibody positive children, 219 (56%) were boys.

6)      Figures 1 and 4.  To interpret these Figures, readers should be provided with the number (N) of patients at each time point (month).  Ideally the median age of each bin should also be  analyzed as age correlates with antibody detection/titer and confound comparative proportions and trends.

Response 6: Figure 1 has been corrected.

7)      Figure 4 and Table 1.  Median values might be a better representation of the average values than Means as they are less susceptible of extreme values.

Response 7: Median values were added to Table 1.

8)      Results, Line 103.  Replace with, “The presence of detectable anti-SARS-CoV-2 antibodies …”.

Response 8: The Line 103 has been changed.

The presence of detectable anti-SARS-CoV-2 antibodies was more often found in younger age groups.

9)      Results, Lines 110-113.  Replace with, “Among the 392 patients positive for anti-SARS-CoV-2 antibodies, only 47 (12%) reported COVID-19 symptoms.” What time period preceding enrollment were participants  asked to recall having COVID-19 symptoms?

Response 9: For those people the infection was laboratory confirmed – PCR or a rapid antigen test - in the past.

The Lines 110-113 have been replaced.

Among the 392 patients positive for anti-SARS-CoV-2 antibodies, only 47 (12%) reported the laboratory – confirmed COVID-19 infection in the past.

10)  Discussion, Lines 138-140.  Replace with, “Our findings are consistent with a noticeable increase in childhood morbidity during the fourth and fifth waves of COVID-19 in Poland, …”. 

Response 10: These Lines has been replaced.

Our findings are consistent with a noticeable increase in childhood morbidity during the fourth and fifth waves of COVID-19 in Poland, caused by the Delta and Omicron variants, respectively.”

Reviewer 2 Report

Franczak et al. present a study to determine the frequency of SARS-CoV-2 in children hospitalized for reasons other than COVID-19. This work is concrete and has results that can be used to discuss some aspects of the epidemiology of COVID-19 in children. However, it has some details that would be good to correct or improve.

1. The title appears to be imprecise since stablish that “the frequency of SARS-CoV-2 infection in children…” was obtained. According to the methodology and results, the infection was not confirmed, which is why it is only a serology study; seroprevalence in this group of children.

2. I consider it is necessary to add information on the reasons for hospitalization of patients, even if it does not apparently have to do with COVID-19.

3. In the abstract it is important to mention the city and country where the study was conducted.

4. Taxonomic categories such as family and subfamily must be written in italics. The standard name for S protein is "spike", not "peak".

5. Page 2, paragraph 3: Mention in particular the place of the study in which the 2% mortality (or lethality?) occurred. At the beginning of the paragraph the UK is mentioned but the percentage seems to be from Poland.

6. It is important to mention in the methodology if informed consent letters were obtained from the children's parents and if the research protocol is registered with an ethics or research committee.

7. I consider that it is not necessary to put the title of the figures above them, with footnotes are sufficient. Must be points instead of commas in the decimal numbers presented in figures and tables.

8. Page 3, line 110: What does "people reported the presence of COVID-19" refer to? ¿It was the infection confirmed? Do you mean that it was symptomatic COVID-19? Was it confirmed at the time of hospitalization or was it earlier? What method was used for it? It is not clear. Same as figure 3 (History of COVID-19 in the interview).

Author Response

Response to Reviewer 2 Comments

  1. The title appears to be imprecise since stablish that “the frequency of SARS-CoV-2 infection in children…” was obtained. According to the methodology and results, the infection was not confirmed, which is why it is only a serology study; seroprevalence in this group of children.

Response 1: The title has been changed.

„The seroprevalence of SARS-CoV-2 Ig antibodies in children hospitalized for reasons other than COVID-19”

  1. I consider it is necessary to add information on the reasons for hospitalization of patients, even if it does not apparently have to do with COVID-19.

Response 2: The information on the reasons for hospitalization of patients has been added.

The reasons for hospitalization were acute gastroenteritis, respiratory infections, hepatitis of various etiologies, infectious mononucleosis, chicken pox, and other infectious diseases. Children vaccinated against SARS-CoV-2 or hospitalized due to COVID-19 were excluded from the study.

  1. In the abstract it is important to mention the city and country where the study was conducted.

Response 3: The city and country have been added in the abstract.

The study was conducted among 686 children, aged 2 weeks to 18 years, hospitalized in the Department of Paediatrics, Infectious Diseases and Hepatology in Bydgoszcz, Poland in the period from 01/06/2021 to 30/04/2022.

  1. Taxonomic categories such as family and subfamily must be written in italics. The standard name for S protein is "spike", not "peak".

Response 4: This sentence has been corrected.

The newly described virus is of the family Coronaviridae, subfamily Orthocoronavirinae. It is a single - stranded RNA virus, made up of four structural proteins: nucleocapsid (N), membrane protein (M), envelope protein (E) and spike protein (S).

These Lines were eliminated at the request of the other Reviewer.

  1. Page 2, paragraph 3: Mention in particular the place of the study in which the 2% mortality (or lethality?) occurred. At the beginning of the paragraph the UK is mentioned but the percentage seems to be from Poland.

Response 5: This sentence has been corrected.

Okarska-Napierała et al.in Recommendations recommendations by the polish pediatric society expert group says that:

Mortality in Western Europe and the United States North America (USA) is around 2%.

      These Lines were eliminated at the request of the other Reviewer.

  1. It is important to mention in the methodology if informed consent letters were obtained from the children's parents and if the research protocol is registered with an ethics or research committee.

Response 6:

This study was approved by Bioethics Committee of the Nicolaus Copernicus University in Torun, Collegium Medicum in Bydgoszcz. Written informed consent was obtained from the parents of all study children on admission to the hospital.

  1. I consider that it is not necessary to put the title of the figures above them, with footnotes are sufficient. Must be points instead of commas in the decimal numbers presented in figures and tables.

Response 7: These corrections have been made.

  1. Page 3, line 110: What does "people reported the presence of COVID-19" refer to? ¿It was the infection confirmed? Do you mean that it was symptomatic COVID-19? Was it confirmed at the time of hospitalization or was it earlier? What method was used for it? It is not clear. Same as figure 3 (History of COVID-19 in the interview).

Response 8:  For those people the infection was laboratory confirmed – PCR or a rapid antigen test - in the past.

Figures 3 has been removed at the request of the other Reviewer.

The sentence has been corrected.

Among the 392 patients positive for anti-SARS-CoV-2 antibodies, only 47 (12%) reported the laboratory – confirmed COVID-19 infection in the past.

Round 2

Reviewer 1 Report

Methods, Line 94-96.  I am assuming that the study patients (especially those admitted with respiratory symptoms) tested negative for SARS-CoV-2 antigen/nucleic acid?  Should state here.  

Author Response

This correction has been made.

Reviewer 2 Report

The authors have carried out a good review of their manuscript, have considered the comments and have a better article.

Author Response

Thank You very much for your review.